# Effects of *Olea europaea* L. Polyphenols on the Animal Welfare and Milk Quality in Dairy Cows

**DOI:** 10.3390/ani13203225

**Published:** 2023-10-15

**Authors:** Maria Chiara Di Meo, Antonia Giacco, Armando Zarrelli, Vittorio Maria Mandrone, Livia D’Angelo, Elena Silvestri, Paolo De Girolamo, Ettore Varricchio

**Affiliations:** 1Department of Sciences and Technologies (DST), University of Sannio, 82100 Benevento, BN, Italy; mardimeo@unisannio.it (M.C.D.M.); agiacco@unisannio.it (A.G.); vittorio.mandrone13@gmail.com (V.M.M.); silves@unisannio.it (E.S.); 2Department of Chemical Sciences, University of Naples Federico II, 80126 Naples, NA, Italy; zarrelli@unina.it; 3Department of Veterinary Medicine and Animal Production, University of Naples Federico II, 80137 Naples, NA, Italy; livia.dangelo@unina.it (L.D.); degirola@unina.it (P.D.G.)

**Keywords:** *Olea europaea* L., polyphenols, animal welfare, milk quality, milk fatty acids, antioxidant, functional feed, dairy cows

## Abstract

**Simple Summary:**

The use of by-products from the olive oil industry in livestock farming is an important resource due to their high content of biomolecules with strong antioxidant properties. In this study, 30 Italian Holstein–Friesian cows were fed with an *Olea europaea* L. polyphenol-enriched diet (500 mg/cow/day) to evaluate the effects of such supplementation on metabolic parameters and milk quality. The enriched diet helped (1) maintain cows’ metabolic parameters in the physiological range producing hypoglycemic and hypolipidemic effects, and (2) improve the fatty acid profile of milk, paralleled by an increase in protein content and lactose. The obtained results show that in lactating dairy cows, supplementation with *Olea europaea* L. extracts can be a valid strategy to improve animal welfare and milk quality.

**Abstract:**

Here, we evaluated the effect of dietary supplementation with an *Olea europaea* L. extract on the animal welfare and milk quality of dairy cows. Thirty Italian Holstein–Friesian dairy cows in the mid-lactation phase (90 to 210 days) were blocked into experimental groups based on parity class (namely, primiparous (P) (n = 10), secondiparous (S) (n = 10) and pluriparous (PL) (n = 10)) and received, for 60 days, Phenofeed Dry^®^ at 500 mg/cow/day. Milk and blood samples were collected before the start of the treatment (T0), subsequently every 15 days (T1–T4) and at 45 days after the end of treatment (T5). In the serum, glucose and triglycerides, stress, the thyroid, lactation and sex hormones were measured; in the milk, lysozyme content as well as the fatty acid profile were assessed. In the whole animal, the enriched feed helped to maintain hormonal parameters in the physiological range while producing hypoglycemic (T4 vs. T0, for P and PL *p* < 0.001) and hypolipidemic effects (T4 vs. T0, for P *p* < 0.001 and for PL *p* < 0.01). At the milk level, it resulted in a reduction in total fat (T5 vs. T0, for P, S and PL *p* < 0.001) and in the saturated fatty acids (SFAs)/monounsaturated fatty acids (MUFAs) ratio paralleled by an increase in polyunsaturated fatty acids (PUFAs) (T5 vs. T0, for P, S and PL *p* < 0.001), protein content (lysozyme (T4 vs. T0, for P and PL *p* < 0.001)) and lactose (T5 vs. T0, for P, S and PL *p* < 0.001). Thus, the inclusion of natural bioactive molecules such as *O. europaea* L. polyphenols in the dairy cow diet may help to improve animal welfare and milk quality.

## 1. Introduction

In recent years, the use of phytocompounds, such as food supplements with beneficial properties and stimulators of animal welfare [1,2], is becoming extensively employed in livestock farming, particularly in intensive farming. Several studies conducted on diets supplemented with biomolecules show the ability of phenolic compounds to improve productive performance and the immune response, and to reduce the oxidative stress and inflammatory status [3,4,5,6]. By preventing the interaction of free radicals with cellular DNA as well as acting on the intestinal microbiota by increasing the digestibility and absorption of nutrients [7], these substances possess antioxidant, immunomodulating/anti-inflammatory and anti-carcinogenic action [6,8,9]. Polyphenols may find new applications in the production of functional foods, which are a valuable aid in reducing the use of antibiotics and drugs currently used in animal breeding [10,11,12]. In recent years, there has been a progressive increase in the use of feed or functional foods, for which, by definition, beyond the basic nutritional properties, their ability to positively influence one or more physiological functions has been scientifically demonstrated. A fundamental prerogative of polyphenols is also to help preserve or improve the health status of humans and animals [1]. A waste/refusal of processing in the agri-food chain representing a disposal cost that affects the general production costs can become a precious economic resource with important repercussions in terms of benefits from health, social-health and environmental aspects [13]. Therefore, supplementing the feed ration with natural antioxidants such as polyphenols may become an optimal alternative to ensure the welfare of dairy cows with a sustainable approach. In organic farms, access to polyphytic pastures, with high plant biodiversity, ensures an appropriate intake of polyphenols [14]. In the case of intensive farming, to obtain such a result, it is necessary to supplement the ration with food supplements enriched with polyphenols. This strategy is supported by several studies conducted in farms in which diets integrated with natural antioxidants have shown an improvement in the quality of animal products [15,16,17]. Among the physio-pathological alterations that may affect dairy cows, there is a growing interest in diseases associated with oxidative stress [18,19], considered, to date, a metabolic disorder affecting all organs and damaging not only the health of animals but also the quality of finished products, such as milk and dairy products [16,20]. The introduction of olive–olive co-products into the diets of farm animals has been shown to improve not only their welfare status but also the shelf life of animal products due to their content of polyunsaturated fatty acids and polyphenols [21] and the quality of animal products such as milk, cheese and eggs [22]. Milk is an important source of SFAs, MUFAs and PUFAs (i.e., omega-3 and omega-6) [23], and the content of the essential fatty acids omega-3 and omega-6 depends on the dietary intake of cows [24]. Moreover, milk is rich in biomolecules, whose quantity is conditioned by breeding techniques and by the welfare status of the cows [25]. Therefore, the evaluation of nutritional composition and the protein content with antibacterial properties in milk becomes essential. Among these, lysozyme, which synergistically works with lactoferrin, has a high antibacterial activity by enhancing the immune response, through mechanisms of nonspecific immunity [26,27]. The concentrations of these proteins in cow milk are extremely low during lactation, at average values of 0.13 µg/mL [28], lower than those of other species such as jenny milk: 1.0 to 3.7 mg/mL [29]; camel: 1.12 µg/mL [30]; ewe: 0.20 µg/mL [31]; and goat: 0.25 µg/mL [32]. Colostrum, but also mastitic milk, contains higher lysozyme concentrations [33,34]. Based on these considerations, the objective of this study was to evaluate the effect of dietary supplementation with an Olea europaea L. extract on the animal welfare and milk quality in the Holstein–Friesian dairy cows.

## 2. Materials and Methods

Animal procedures were reviewed and approved by the Ethical Animal Care and Use Committee of the University of Naples “Federico II” (Protocol No. 99607-2017).

### 2.1. Animals and Experimental Design

This study was carried out at the dairy farm “Fratelli Mirra” located in Francolise (CE) (Italy) (41°07′56″ N 14°03′43″ E; 10 m above sea level) on thirty Italian Holstein–Friesian dairy cows in mid-lactation phase (149 ± 42 days in milk), blocked into three experimental groups based on parity class (primiparous (P) (n = 10) (body weight, 580 ± 20 kg), secondiparous (S) (n = 10) (body weight, 630 ± 10 kg) and pluriparous (PL) (n = 10) (body weight, 640 ± 15 kg)) [35]. This study lasted 105 days. Cows were fed with a total mixed ration (TMR) and oat hay and milked twice daily in the morning and afternoon in the milking parlor. Animals were selected and divided into the experimental groups according to days in milk and parity; for 60 days, cows received a supplement with *Olea europaea* L. phenolic extract (Phenofeed Dry^®^) (500 mg/cow/day) added in powder form into the feed ration of dairy cows. Milk and blood samples were collected before the start of the administration of the functional feed, at time 0 (control time, T0); during supplementation at 15 (T1), 30 (T2), 45 (T3) and 60 (T4) days; and at 45 days after the end of treatment (follow-up, T5) [35].

### 2.2. Diet Composition and Functional Feed

Standard diet composition, chemical–nutritional analysis, functional analysis (characterization of total phenolic content and antioxidant activity) and fatty acid profile of the feed enriched with Phenofeed Dry^®^ extract are reported in Di Meo et al. [35].

### 2.3. Bovine Blood Collection and Analysis

Blood samples from each individual cow were collected every 15 days before and after milking, as reported by Di Meo et al. [35]. Plasma samples were used for fatty acids analysis [35], while serum samples were used for metabolic analysis, as described below.

#### 2.3.1. Serum Concentrations of Glucose and Triglycerides

Glucose and triglyceride concentrations in bovine serum were determined through enzymatic colorimetric methods using commercial kits (Giesse Diagnostics, Colle Prenestino, Rome, Italy). The rates for individual cows, at the different sampling times, were analyzed using 1000 µL of reagent and 10 µL of sample. The blank and the standard were initially prepared in duplicate with the addition of 1000 µL of reagent and 10 µL of distilled water for the blank and 1000 µL of reagent and 10 µL of standard. The samples were shaken and incubated at 37 °C for 10 min for glucose determination and 5 min for triglycerides; subsequently, the reading was carried out in duplicate on the spectrophotometer (Biomate 3) at wavelengths of 510 and 546 nm for glucose and triglycerides, respectively. Results are expressed as mg/dL of bovine serum.

#### 2.3.2. Serum Levels of Hormonal Parameters

Specific Enzyme-Linked Immunosorbent Assay (ELISA) kits (Diametra, Foligno, Italy) were used to assess serum concentrations of free triiodothyronine (fT3), thyroxine (fT4) and 17β-estradiol. Analyses were performed in duplicate. Results are expressed as ng/dL for fT4 and as pg/mL for fT3 and for 17β-estradiol. Serum levels of oxytocin and prolactin were measured to evaluate putative effects of the functional food on lactation. Serum levels of adrenocorticotropic hormone (ACTH) and cortisol were measured as markers of stress and oxidative imbalance. Oxytocin levels were measured using an ELISA kit from DRG Diagnostic, Germany, and prolactin, ACTH and cortisol were measured using specific ELISA kits from FineTest, Wuhan, China.

### 2.4. Milk Sampling and Analysis

Milk was collected from each cow every 15 days for 60 days and at 45 days after the end of treatment (follow-up, T5), immediately transferred to the laboratory, divided into several aliquots and stored at −20 °C until used for analyses. About 10 mL of milk was used for fatty acid composition analysis, while other aliquots were used to perform the ELISA Lysozyme Food assay (DRG Diagnostic, Marburg, Germany, EIA-6027) for the quantitative determination of lysozyme in milk.

#### 2.4.1. Analysis of Fatty Acid Profile

To analyze fatty acid composition, milk samples were extracted with methylene chloride and the organic phase was dried under a slight flow of nitrogen. As previously reported [36], each sample was dried for 1 h over P_2_O_5_ and then treated for 20 min at 60 °C with a solution of boron trifluoride/methanol 10% (1.3 M, 0.5 mL) and 100 µL of dimethoxypropane. The crude mixture of fatty acid methyl esters thus obtained was extracted twice with hexane and the organic phases were combined and dried. Gas chromatography (GC-FID) analyses were obtained on a Shimadzu model GC2010 instrument equipped with an SP52–60 capillary column (Sigma-Aldrich, St Louis, MO, USA; 100 m × 0.25 inside diameter × 0.20 film thickness) and performed as reported in Di Meo et al. [35].

#### 2.4.2. Lysozyme Content and Chemical–Nutritional Analysis of Milk

For milk quality analysis, the lysozyme levels were determined by using a specific commercial ELISA kit, according to the manufacturer’s instructions, and the absorbance at 450 nm was determined using a Biomate 3-Thermo Spectronic spectrophotometer (Thermo Fisher, Waltham, MA, USA). Results are expressed as ppb and the measurements were performed in duplicate.

The chemical composition of milk (total fat, protein and lactose) and hygiene parameters (somatic cell count (SSC)) were determined using the Speedylab analyzer (Astori Tecnica snc, Brescia, Italy) based on a sensitive and specific ultrasonic technique, and the Lactoscan Milk Analyzer (Bulgaria), respectively. All analyses were carried out in duplicate.

### 2.5. Statistical Analysis

Before statistical analysis, the data were tested for normality using the Shapiro–Wilk test [37]. For comparisons between two groups, statistical significance was calculated using non-paired two-tailed Student’s *t*-test. For multiple comparisons, One-way ANOVA (post hoc test: Student–Newman–Keuls) was performed. Graphs and calculation of statistical significance were performed using Graph Pad Prism 8 software (GraphPad, San Diego, CA, USA). Bars are represented as the standard deviation (SD) of the mean. For all analyses, *p* value < 0.05 was considered the minimum statistical significance. Differences between cows for each parity class (P, S, PL) were analyzed using One-way ANOVA statistical analysis (Newman–Keuls post-test) each time, while two-way ANOVA (Tukey test) was used to assess the effects of dietary treatment in each parity class at different experimental points (T0, T1, T2, T3, T4, T5). The experimental results (T1-T5) were compared with the control (T0) to evaluate the blood parameters and milk composition, during and after the experimental phase.

For simplicity in reading the data, it was decided to report the One-way ANOVA statistical analysis in graphs and tables, while the Two-way ANOVA analysis was reported in the legend for each analyzed parameter. Only for the table of fatty acids of cow milk (Table 1), the Two-way ANOVA is reported in Appendix A.

Moreover, to make the data reported in Figures 1–8 more understandable in terms of actual change, a strategic table containing mean values and One-way ANOVA results is available in Appendix A (see Appendix A).

## 3. Results

### 3.1. Serum Glucose Levels in P, S and PL Cows upon Polyphenol-Enriched Diet

Figure 1 shows the time-course analysis (T0–T5) of serum glucose levels measured in P, S and PL cows during the two months of treatment. In all three experimental groups (P, S, PL), the polyphenol-enriched feed determines a hypoglycemic effect, more evident within the first month (T2), while it fades at the second month (T4). On average, the values decrease from T0 over the treatment period until the T5 (follow-up), corresponding to the end of lactation phase.

### 3.2. Serum Levels of Triglycerides in P, S and PL Cows upon Polyphenol-Enriched Diet

As reported in Figure 2, in all three experimental groups, the enriched feed produces a statistically significant reduction (*p* < 0.001) in the serum levels of triglycerides. In the time interval T2–T4, the hypotriglyceridemic effect is, on average, more evident in the P group. In this group, the values reduce to one third compared to the controls at T3 (T3 vs. T0), whereas, in the S and PL groups, at the same time point, the values are halved, suggesting a treatment effect dependent on parity class. At the follow-up, 45 days after the last administration of enriched feed, in all three experimental groups, the triglyceride levels reach significantly higher values than those of the control time (T0).

### 3.3. Serum Levels of fT3 and fT4 in P, S and PL Cows upon Polyphenol-Enriched Diet

As shown in Figure 3, in all three experimental groups, serum levels of fT3 and fT4 (Figure 3A,B, respectively) do not appear to be affected by the dietary regimen, remaining at values within the euthyroid range throughout the observation period.

### 3.4. Serum Levels of 17β-Estradiol in P, S and PL Cows upon Polyphenol-Enriched Diet

As shown in Figure 4, serum levels of 17β-Estradiol remain unchanged in cows of P and S groups over the entire treatment period. Interestingly, PL cows show periodic peaks with a statistically significant increase at monthly frequency (T2 and T4). At T5, all cows show increased values of serum 17β-Estradiol.

### 3.5. Serum Levels of ACTH and Cortisol in P, S and PL Cows upon Polyphenol-Enriched Diet

As shown in Figure 5, serum levels of ACTH are significantly increased in a parity-class-dependent manner, with the highest values in PL cows compared to S and P cows.

In parallel, cortisol levels, although not statistically significant, show an increase dependent on parity class during treatment, with higher values in PL cows (Figure 6).

### 3.6. Serum Levels of Prolactin and Oxytocin in P, S and PL Cows upon Polyphenol-Enriched Diet

In all experimental groups (P, S and PL), prolactin levels remain constant during the feeding treatment (Figure 7). As far as the serum levels of oxytocin are concerned, no variations between groups and time points are observed (Figure 8).

### 3.7. Fatty Acids Profile of Milk from P, S and PL Cows upon Polyphenol-Enriched Diet

Table 1 shows the content of saturated (SFAs) and unsaturated (MUFAs and PUFAs) fatty acids in the milk of the dairy cows during the experimental period.

Milk FA composition is strongly influenced by polyphenol supplementation (Table 1 and Appendix A in Appendix A). Total SFA, MUFA and PUFA contents change significantly during the experimental periods: the total content of SFA decreases in the polyphenol-supplemented diet animal in favor of the MUFA and PUFA concentrations in all parity classes (*p* < 0.001), with a statistically significant increase even after the end of the treatment (T5). Indeed, the enriched diet contributes to a lower SFA/MUFA ratio in cow milk. As expected, at time T0, the SFA/MUFA ratio is higher in S and PL compared to P (*p* < 0.001); on the contrary, at the end, and during the follow-up, the differences between the groups are comparable.

Regarding the PUFA ratio omega-6/omega-3, before the supplemented diet administration, cows show a significantly higher content of omega-6 than omega-3 in the milk. During the treatment, a significant increase in both fatty acids is detected already after one month of diet administration (*p* < 0.001), and the ratio is conserved in the optimal range throughout the experimental period.

### 3.8. Lysozyme Content and Chemical–Nutritional Analysis in Milk from P, S and PL Cows upon Polyphenol-Enriched Diet

Figure 9 shows that, in all the experimental groups (P, S and PL), the lysozyme content in cow milk significantly increases during the dietary treatment (*p* < 0.001). Lower values are observed at T0 and T5. Compared to control time (T0), a higher lysozyme content (*p* < 0.001) persists in S and PL cows at time T5, likely indicating a long-lasting antibacterial effect of diet polyphenols in such groups. At each time point, there are very few differences between the parity classes. Concerning the different parity classes, significant differences are found at time T0 (*p* < 0.01) and T5 (*p* < 0.01; *p* < 0.05), but not during the dietary treatment. The chemical–nutritional composition of milk (protein, lactose and fat content) and the somatic cell content are reported in Table 2. While no significant differences are observed between the cows of different parity classes, an increased content of lactose and protein is observed when comparing T0 and T5. In addition, significant differences are observed only in the somatic cell content, which progressively lowers during the whole treatment and remains stable after the end of treatment (T5).

## 4. Discussion

The addition of phenolic compounds derived from products and by-products of the main agri-food supply chains in the diet of lactating cows represents an important resource [38]. The large production quantities, environmental impact and nutritional content of by-products from the olive oil industry make them an important subject for an accurate valorization and implementation in animal nutrition [3]. In this regard, our study aims, with a view to the circular economy, to implement co-products of the olive oil sector in the livestock sector as an important biomolecule resource, given the high production of waste/by-products in the Mediterranean area. Several studies show that the inclusion of these by-products/co-products in the diets of monogastric and ruminants should be monitored due to the high fiber and fat content in the feed ration [21,39], emphasizing that moderate intake does not affect the growth performance of animals, but may improve the fatty acid profile of animal products by reducing the share of saturated fatty acids and increasing unsaturated ones [40].

Our findings show a significant effect of the enriched diet on the glucose and lipid metabolism of the cows, and on the milk fatty acid profile, while the hormonal status (stress hormones and hormones involved in lactation) is maintained within the physiological range, indicating a lack of putative adverse effects of polyphenolic compounds on specific endocrine axes.

There are several studies that also report the antioxidant effects of other types of extracts (pomegranate peel [38] and dried pomace [41]) when added to the diet of dairy cows, but which did not lead to significant differences in serum glucose and triglyceride levels. In contrast, goats fed tea catechins showed an increase in metabolic parameters, particularly glucose content [38,42,43]. Thus, we can hypothesize that the hypoglycemic and hypotriglyceridemic effect observed in our study may be related to the animal species, type and/or concentration of *O. europaea* polyphenols combined with the diet.

Thyroid hormones, whose serum levels are influenced by the lactation phase, play an important role in determining the intensity of cell metabolism, and are known to affect the metabolism of lipids and carbohydrates, the main constituents of the ruminants’ diet [44]. A positive correlation between the circulating levels of thyroid hormones, fT4 and fT3, on the one hand and of glucose and lipids on the other has been reported [45]. In our study, the serum levels of fT4 and fT3 are within the euthyroid range in all experimental groups, suggesting a protective effect of the diet against the risk of developing hypothyroidism, a condition leading to oxidative stress and altered blood glucose and lipid metabolism [46].

There are few studies on the effects of diet on 17β-estradiol levels, a steroid hormone whose elevated secretion drastically reduces milk production, in dairy cows. According to our data, the observed fluctuations of these hormones are more likely influenced by the physiological condition of the cow (i.e., lactation, dry period, pregnancy, estrus cycle, transition period) [47].

Still on the hormonal status, cortisol and ACTH levels in dairy cows can be used as indices of physiological or environmental stress, with consequences on the welfare and quality of cow production [48]. Here, we find that the enriched diet does not influence cortisol and ACTH levels, as well as the milking procedures (i.e., prolactin and oxytocin). However, in our study, significant differences in the levels of cortisol and ACTH are observed when comparing the parity classes. In particular, the pluriparous cows show higher basal levels of cortisol and ACTH associated with a general condition of chronic stress probably due to the higher calving number [49]. The experimental dietary treatment, independently of the parity class, also contributed to keeping serum prolactin levels stable. Sani et al. [50] report that in rats, a diet enriched with *Launaea taraxacifolia* and resveratrol leads to an increased serum concentration of prolactin and oxytocin due to the increased secretory activity of the mammary glandular alveoli, resulting in increased milk secretion.

In ruminants, milk fatty acids are primarily derived from the diet, the de novo synthesis in the mammary gland and the production by ruminal bacteria [51]. Palladino et al. [52] report that the diet is among the main factors influencing the fatty acid content of cow milk, particularly PUFAs. Ruminant milk is predominantly rich in SFAs (on average 60–70% of total fatty acids) than in unsaturated ones (MUFAs and PUFAs), and the ratio between SFAs, MUFAs and PUFAs has been reported to be primarily influenced by the lactation phase and the diet composition [53,54]. In general terms, ruminant milk fat is poor in PUFAs such as linoleic acid (C18:2, ω-6) and α-linolenic acid (C18:3, ω-3). The main goal of our research was to verify whether the experimental polyphenol-enriched diet could significantly modify the acid profile of the milk produced by the cows under treatment. Independently of the parity class, the administration of the functional feed determines a significant reduction in the SFA content in favor of those of MUFA and PUFA in milk. This effect is also evident in P animals whose milk, at T0, was characterized by a high concentration of PUFA (49.9%). It should be noted that our study confirms that cow milk is the richest source of oleic acid (24%) (MUFA) compared to the milk of other animal species (goats, sheep), composed of 18% of total fatty acids on average, corroborating the high functional value of cow milk [55].

We hypothesize that the fatty acid diet composition and its enrichment with bioactive molecules influence the content of fatty acids in milk. Cimmino et al. [17] showed that in goats, the supplementation with 3.2 mg/day of powdered polyphenols extract derived from olive mill wastewater resulted in a significant reduction in short-chain fatty acids with a higher proportion of monounsaturated fatty acids in the derived meat. This suggests the possible influence of polyphenols on the enzyme desaturase Δ-9, the key enzyme required to convert palmitic to palmitoleic acid and stearic to oleic acid. Such a mechanism could also be involved in the increase in PUFA observed in the milk produced by the cows analyzed in our study.

Moreover, several other studies show a significant reduction in the SFA/MUFA ratio following the addition of phenolic compounds to the diet also in other livestock (buffaloes, goats, sheep). In this regard, Terramoccia et al. [56], Vargas-Bello-Pérez et al. [57] and Chiofalo et al. [58] point out that the addition of 15–25% of olive cake to the diet results in an increase in C18:1n-9 (MUFA) in ruminant milk (buffalo and ewe). Malla et al. [59] report in other animal species (goat, sheep, camel) a higher SFA/MUFA ratio (2.42% in goat, 2.70% in sheep, 2.24% in camel).

We underline the good “functionality” of our dietary treatment considering that the milk of the experimental cows was also enriched in lysozyme content. Lysozyme is an enzymatic protein functioning in synergism with immunoglobulins and lactoferrin, showing antibacterial activity against a wide variety of bacteria [26]. Krol et al. [26] found high levels of lysozyme and immunoglobulins in bovine milk as lactation progressed. In our study, higher values of milk lysozyme content are detected (>20 µg/L) compared to those reported in the literature. Specifically, Krol et al. [26] show that the lysozyme content, in the mid-lactation phase, is 9.18 µg/L and is lower in primiparous and higher in pluriparous cows, while Reklewska et al. [60] show a lysozyme content of 12.54–16.43 μg/L mainly influenced by the diet and season (higher in summer). It is well known that the content of this protein is very low in bovine milk [27,28].

To our knowledge, this is a pilot study. Although this study provides promising new perspectives on the use of the functional feed on dairy cow welfare and milk composition, the main concern may be about the sample size. The limited sample number is due to the low availability of PhenoFeed Dry^®^ in the dose we tested to be effective. Therefore, even though more samples were initially enrolled, some animals unfortunately experienced an early dry phase.

## 5. Conclusions

In conclusion, this study shows that the polyphenol-enriched diet of *Olea europaea* L., in Holstein–Friesian cows in the mid-lactation phase, influences glucose and lipid metabolism and the fatty acid composition of milk while ensuring hormonal physiology and balance and, thus, a general state of wellbeing in the experimental cows. The enriched diet with olive extract PhenoFeed Dry^®^ appears effective in (*i)* promoting the reduction in SFAs, (*ii)* increasing MUFAs and PUFAs and (*iii)* ensuring an adequate protein content in the milk, in particular, lysozyme, with known antibacterial properties.

Our results show that *O. europaea* L. polyphenols contribute to improving the quality of cow milk and represent an important resource, also in view of the circular economy, to produce feeds with a high functional value for dairy cows. Finally, in the context of the existing literature on the use of *O. europaea* extracts as a supplement for cattle diets, the advancement provided by our study consists of having tested a standardized, highly reproducible, functional extract (PhenoFeed Dry^®^), which, compared to other enrichment methods, may represent an exemplary zootechnical approach.

### Future Implications

This study reinforces the hypothesis that supplementing the diet of animals in livestock production with bioactive molecules from *O. europaea* may be an interesting strategy from a feedstuff standpoint to replace and/or reduce part of the OGM raw material and synthetic supplements. The use of natural antioxidant molecules in the olive oil supply chain contributes to ensuring welfare conditions in animals under production stress, and to improving the nutritional and functional quality of food of animal origin.

The recent diffusion of functional feeds and recent insights on the physiological effects in livestock and production quality have stimulated the interest of the scientific community in broadening knowledge about the effects of animal-derived products in humans, an advantage that helps with the beneficial use of using sustainable/disposable products in the perspective of the circular economy.

The results of this study give further insights into the discussion, started by many researchers, aiming to improve the fatty acid composition of cow milk according to the diet, production process, breed and lactation phase of the cows. However, further studies are needed to further characterize the effects and efficacy of the diet supplemented with *O. europaea* polyphenols on the rheological properties and lipid composition of milk and milk products. In this regard, administration of this diet to lactating cows in a whole herd and/or to cows in different physiological phases, such as the dry phase and the transition phase, may help to better characterize the beneficial effect on reproductive performance.

## Figures and Tables

**Figure 1 animals-13-03225-f001:**
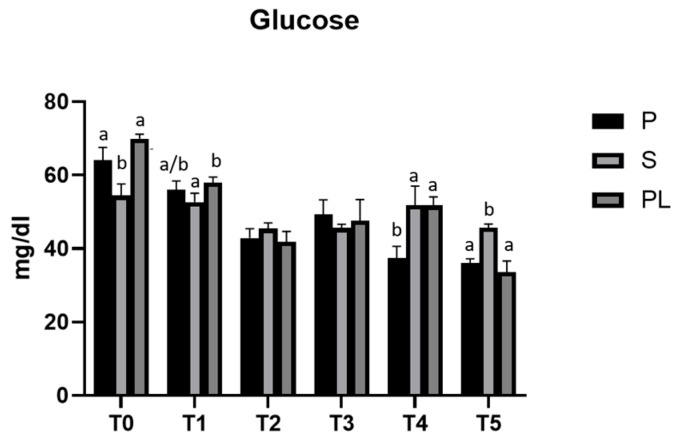
Serum glucose levels in the Primiparous (P), Secondiparous (S) and Pluriparous (PL) cow groups at time T0, T1, T2, T3, T4 and T5 (follow-up). Data are reported as mean ± SD. One-way ANOVA (Newman–Keuls test). a,b is the statistical significance (One-way ANOVA): T0 (S vs. P, PL ** *p* < 0.01); T1 (S vs. PL * *p* < 0.05); T4 (P vs. S, PL ** *p* < 0.01); T5 (S vs. P, PL *** *p* < 0.001). Two-way ANOVA (Tukey test): P (T0 vs. T1 * *p* < 0.05; T0 vs. T2, T3, T4, T5 *** *p* < 0.001; T1 vs. T3 * *p* < 0.05; T2 vs. T5 * *p* < 0.05; T3 vs. T4, T5 *** *p* < 0.001). S (T0 vs. T2 *** *p* < 0.001; T0 vs. T3, T5 ** *p* < 0.01; T1 vs. T2, T3, T5 * *p* < 0.05; T4 vs. T2, T3, T5 * *p* < 0.05). PL (T0 vs. T1, T2, T3, T4, T5 *** *p* < 0.001; T1 vs. T2, T3, T5 *** *p* < 0.001; T1 vs. T4 * *p* < 0.05; T3 vs. T5 * *p* < 0.05; T4 vs. T2, T5 *** *p* < 0.001).

**Figure 2 animals-13-03225-f002:**
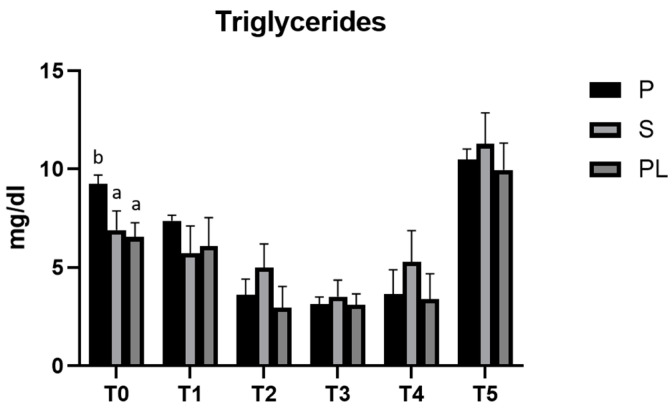
Serum triglycerides levels in the Primiparous (P), Secondiparous (S) and Pluriparous (PL) cow groups at time T0, T1, T2, T3, T4 and T5 (follow-up). Data are reported as mean ± SD. a,b,c is the statistical significance One-way ANOVA (Newman–Keuls test): T0 (P vs. S, PL * *p* < 0.05). Two-way ANOVA (Tukey test): P (T0, T1 vs. T2, T3, T4 *** *p* <0.001; T1 vs. T5 ** *p* < 0.01; T2, T3, T4 vs. T5 *** *p* < 0.001). S (T0 vs. T3, T5 *** *p* < 0.001; T1, T2, T3, T4 vs. T5 *** *p* < 0.001). PL (T0 vs. T2, T3, T5 *** *p* < 0.001; T0 vs. T4 ** *p* < 0.01; T1 vs. T2, T3, T4 ** *p* < 0.01; T1, T2, T3, T4 vs. T5 *** *p* < 0.001).

**Figure 3 animals-13-03225-f003:**
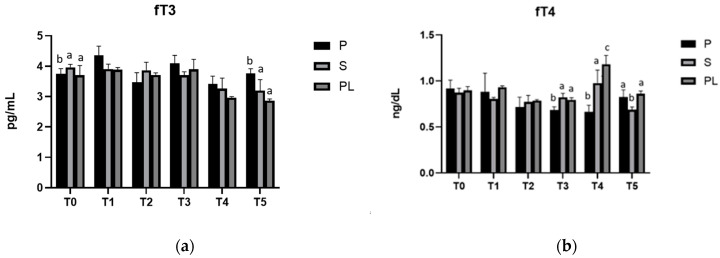
Serum levels of fT3 (**a**) and fT4 (**b**) in the Primiparous (P), Secondiparous (S) and Pluriparous (PL) cow groups at time T0, T1, T2, T3, T4 and T5 (follow-up). Data are reported as mean ± SD. In (**a**), a,b is the statistical significance One-way ANOVA (Newman–Keuls test): T0 (P vs. S, PL * *p* < 0.05). T5 (P vs. S ** *p* < 0.01; P vs. PL *** *p* < 0.001). In Two-way ANOVA test: P (T0 vs. T1 ** *p* < 0.01; T1 vs. T2, T4 *** *p* < 0.001; T1 vs. T5 ** *p* < 0.01; T2 vs. T3 ** *p* < 0.01; T3 vs. T4 ** *p* < 0.01). S (T0, T1, T2 vs. T4 ** *p* < 0.01; T0, T1 vs. T5 *** *p* < 0.001; T2 vs. T5 ** *p* < 0.01; T3 vs. T5 * *p* < 0.05). PL (T0, T1, T2, T3 vs. T4, T5 *** *p* < 0.001). In (**b**), a,b,c is the statistical significance One-way ANOVA (Newman–Keuls test): T3 (*p* vs. S *** *p* < 0.001; P vs. PL ** *p* < 0.01). T4 (P vs. S ** *p* < 0.01; P vs. PL *** *p* < 0.001; S vs. PL * *p* < 0.05). T5 (S vs. P, PL ** *p* < 0.01). Two-way ANOVA test: P (T0 vs. T2, T3 ** *p* < 0.01; T0 vs. T4 *** *p* < 0.001; T1 vs. T2, T3 *** *p* < 0.05; T1 vs. T4 ** *p* < 0.01). S (T0 vs. T5 * *p* < 0.05; T1 vs. T4 * *p* < 0.05; T2 vs. T4 ** *p* < 0.01; T4 vs. T5 *** *p* < 0.001). PL (T0, T1, T2, T3, T5 vs. T4 *** *p* < 0.001).

**Figure 4 animals-13-03225-f004:**
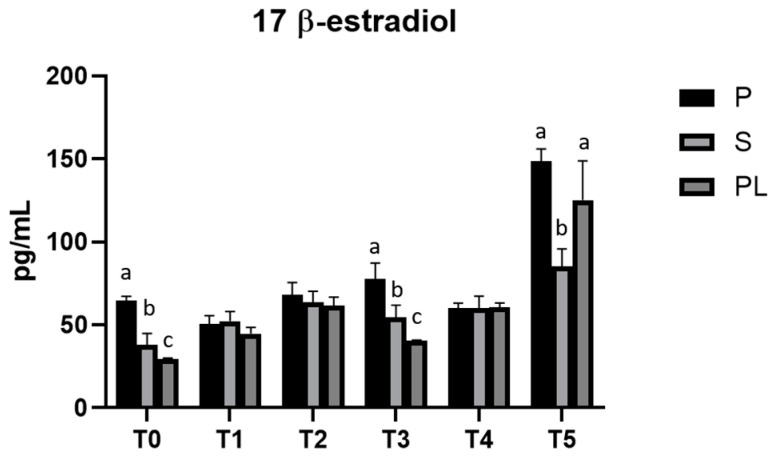
Serum levels of 17-β-Estradiol in the Primiparous (P), Secondiparous (S) and Pluriparous (PL) cow groups at time T0, T1, T2, T3, T4 and T5 (follow-up). Data are reported as mean ± SD. a,b,c is the statistical significance One-way ANOVA (Newman–Keuls test): T0 (P vs. S, PL *** *p* < 0.001; S vs. PL * *p* < 0.05). T3 (P vs. S ** *p* < 0.01; P vs. PL *** *p* < 0.001; S vs. PL * *p* < 0.05). T5 (S vs. P *** *p* < 0.001; S vs. PL ** *p* < 0.01). Two-way ANOVA (Tukey test): P (T0, T1, T2, T3, T4 vs. T5 *** *p* < 0.001; T1 vs. T2 * *p* < 0.05; T1 vs. T3 *** *p* < 0.001; T3 vs. T4 * *p* < 0.05). S (T0 vs. T2, T5 *** *p* < 0.001; T0 vs. T4 ** *p* < 0.01; T1, T3, T4 vs. T5 *** *p* < 0.001; T2 vs. T5 ** *p* < 0.01). PL (T0 vs. T2, T4, T5 *** *p* < 0.001; T1 vs. T2 * *p* < 0.05; T1, T2, T3, T4 vs. T5 *** *p* < 0.001; T2 vs. T3 ** *p* < 0.05).

**Figure 5 animals-13-03225-f005:**
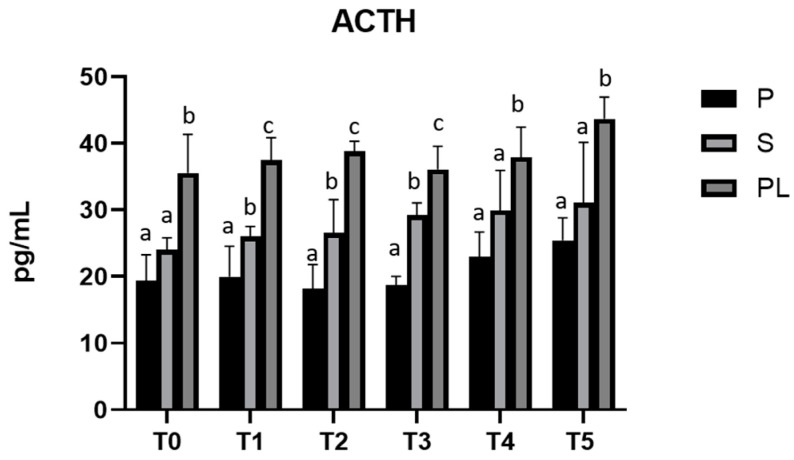
Serum levels of ACTH in the Primiparous (P), Secondiparous (S) and Pluriparous (PL) cow groups at time T0, T1, T2, T3, T4 and T5 (follow-up). Data are reported as mean ± SD. a,b,c is the statistical significance One-way ANOVA (Newman–Keuls test): T0 (PL vs. P, S ** *p* < 0.01). T1 (P vs. S * *p* < 0.05; P vs. PL *** *p* < 0.001; PL vs. S ** *p* < 0.01). T2 (P vs. S * *p* < 0.05; P vs. PL *** *p* < 0.001; PL vs. S ** *p* < 0.01). T3 (P vs. S, PL *** *p* < 0.001; PL vs. S ** *p* < 0.01). T4 (P vs. PL ** *p* < 0.01; PL vs. S * *p* < 0.05). T5 (P vs. PL ** *p* < 0.01; PL vs. S * *p* < 0.05).

**Figure 6 animals-13-03225-f006:**
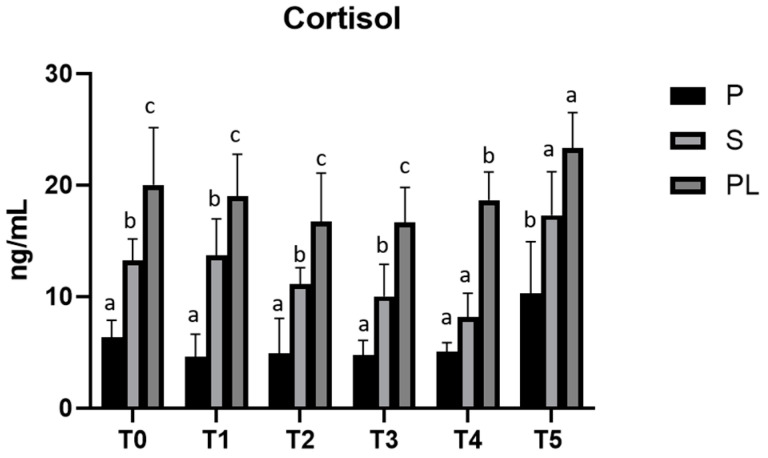
Serum levels of cortisol in the Primiparous (P), Secondiparous (S) and Pluriparous (PL) cow groups at time T0, T1, T2, T3, T4 and T5 (follow-up). Data are reported as mean ± SD. a,b,c is the statistical significance One-way ANOVA (Newman–Keuls test): T0 (S vs. P, PL * *p* < 0.05; PL vs. P *** *p* < 0.001). T1 (P vs. S ** *p* < 0.01; P vs. PL *** *p* < 0.001; PL vs. S * *p* < 0.05). T2 (S vs. P, PL * *p* < 0.05; PL vs. P ** *p* < 0.01). T3 (P vs. S * *p* < 0.05; P, S vs. PL *** *p* < 0.001). T4 (PL vs. P, S *** *p* < 0.001). T5 (P vs. S * *p* < 0.05; PL vs. P ** *p* < 0.01). Two-way statistical analysis ANOVA (Tukey test): S (T3, T4 vs. T5 ** *p* < 0.01). PL (T2, T3 vs. T5 * *p* < 0.05).

**Figure 7 animals-13-03225-f007:**
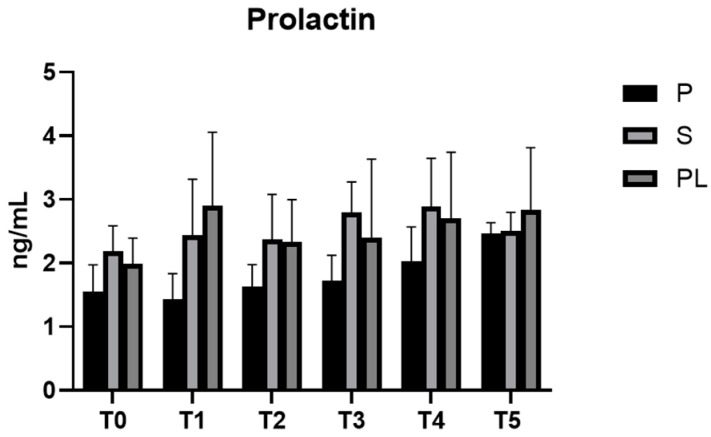
Serum levels of prolactin in the Primiparous (P), Secondiparous (S) and Pluriparous (PL) cow groups at time T0, T1, T2, T3, T4 and T5 (follow-up). Data are reported as mean ± SD.

**Figure 8 animals-13-03225-f008:**
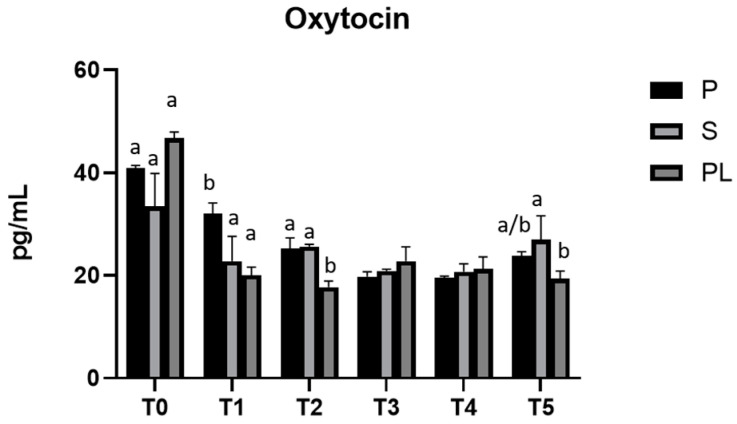
Serum levels of oxytocin in the Primiparous (P), Secondiparous (S) and Pluriparous (PL) cow groups at time T0, T1, T2, T3, T4 and T5 (follow-up). Data are reported as mean ± SD. a,b is the statistical significance One-way ANOVA statistical analysis (Newman–Keuls test): T0 (PL vs. S, P *** *p* < 0.001). T1 (P vs. S, PL ** *p* < 0.01). T2 (PL vs. P, S *** *p* < 0.001). T5 (S vs. PL * *p* < 0.05). Two-way ANOVA statistical analysis (Tukey test): P (T0 vs. T1, T2, T3, T4, T5 *** *p* < 0.001; T1 vs. T2 * *p* < 0.05; T1 vs. T3, T4, T5 ** *p* < 0.01). S (T0 vs. T1, T3, T4 *** *p* < 0.001; T0 vs. T2 ** *p* < 0.01; T0, T3, T4 vs. T5 * *p* < 0.05). PL (T0 vs. T1, T2, T3, T4, T5 *** *p* < 0.001).

**Figure 9 animals-13-03225-f009:**
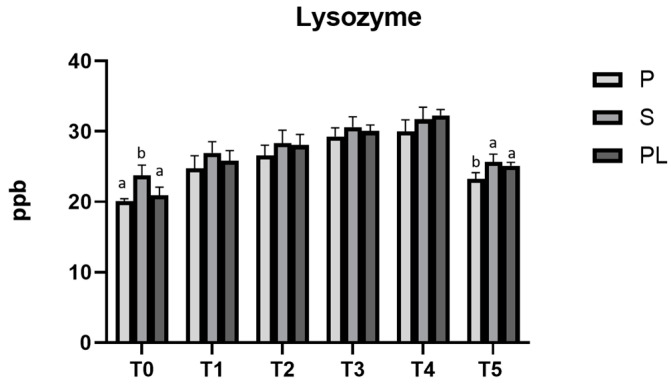
Lysozyme levels in milk from Primiparous (P), Secondiparous (S) and Pluriparous (PL) cows at time T0, T1, T2, T3, T4 and T5 (follow-up). Data are reported as mean ± SD. a,b,c is the statistical significance One-way ANOVA analysis (Newman–Keuls test): T0 (S vs. P, PL ** *p* < 0.01); T5 (P vs. S ** *p* < 0.01; P vs. PL * *p* < 0.05). Two-way ANOVA analysis (Tukey Test): P (T0 vs. T1, T2, T3, T4 *** *p* < 0.001; T0 vs. T5 * *p* < 0.05; T1 vs. T3, T4 *** *p* < 0.001; T2 vs. T4 ** *p* < 0.01, T2 vs. T5 * *p* < 0.05; T5 vs. T3, T4 *** *p* < 0.001); S (T0 vs. T1 * *p* < 0.05; T0 vs. T2, T3, T4 *** *p* < 0.001; T1 vs. T3 ** *p* < 0.01; T1 vs. T4 *** *p* < 0.001; T2 vs. T4 * *p* < 0.05; T5 vs. T3, T4 *** *p* < 0.001); PL (T0 vs. T1, T2, T3, T4, T5 *** *p* < 0.001; T1 vs. T3, T4 *** *p* < 0.001; T2 vs. T4 *** *p* < 0.001; T2 vs. T5 * *p* < 0.05; T5 vs. T3, T4 *** *p* < 0.001).

**Table 1 animals-13-03225-t001:** Fatty acid composition of milk from P, S and PL cows fed with a standard diet (T0) and the Phenofeed Dry^®^ supplemented diet (T1-T5) ^1^.

Fatty Acid (g/100 g)	Experimental Time																	
	T0			T1			T2			T3			T4			T5		
	P	S	PL	P	S	PL	P	S	PL	P	S	PL	P	S	PL	P	S	PL
C14:0	1.09 ± 0.04 ^a^	16.19 ± 0.25 ^b^	15.35 ± 0.33 ^c^	1.00 ± 0.03	14.87 ± 0.29	13.35 ± 0.29	0.90 ± 0.02	14.23 ± 0.32	12.99 ± 0.20	0.80 ± 0.09	12.18 ± 0.10	10.76 ± 0.13	0.68 ± 0.10	9.70 ± 0.18	8.09 ± 0.16	0.60 ± 0.09	9.15 ± 0.17	7.10 ± 0.13
C16:0	15.31 ± 0.33 ^a^	47.50 ± 0.62 ^b^	38.77 ± 0.67 ^c^	15.00 ± 0.28	33.76 ± 0.82	29.28 ± 0.68	14.70 ± 0.20	30.39 ± 0.76	28.79 ± 0.14	13.41 ± 0.13	28.00 ± 0.20	27.03 ± 0.18	12.09 ± 0.10	27.33 ± 0.18	25.18 ± 0.17	11.18 ± 0.11	27.10 ± 0.17	24.40 ± 0.16
C16:1 n7t	4.74 ± 0.05 ^b^	n.d.^a^	n.d.^a^	5.08 ± 0.04	0.10 ± 0.02	0.03 ± 0.01	5.37 ± 0.05	0.22 ± 0.05	0.07 ± 0.01	6.01 ± 0.16	0.56 ± 0.11	0.26 ± 0.09	6.23 ± 0.13	0.86 ± 0.08	0.40 ± 0.04	6.90 ± 0.11	0.90 ± 0.09	0.55 ± 0.04
C16:1 n7	n.d.^a^	0.85 ± 0.02 ^b^	n.d.^a^	0.10 ± 0.02	0.98 ± 0.03	0.05 ± 0.02	0.13 ± 0.03	1.56 ± 0.10	0.10 ± 0.02	0.16 ± 0.06	2.91 ± 0.11	0.25 ± 0.07	0.20 ± 0.06	3.56 ± 0.10	0.42 ± 0.02	0.25 ± 0.05	4.02 ± 0.10	0.58 ± 0.05
C18:0	8.45 ± 0.17 ^a^	9.89 ± 0.13 ^b^	14.43 ± 0.26 ^c^	7.64 ± 0.16	6.24 ± 0.17	10.22 ± 0.19	7.43 ± 0.20	5.77 ± 0.10	9.81 ± 0.11	6.15 ± 0.09	5.09 ± 0.08	8.56 ± 0.11	5.48 ± 0.11	4.39 ± 0.09	7.98 ± 0.08	5.00 ± 0.10	3.06 ± 0.08	6.17 ± 0.12
C18:1 n9	9.36 ± 0.18 ^a^	21.01 ± 0.42 ^b^	26.38 ± 0.53 ^c^	9.60 ± 0.17	24.14 ± 0.34	29.00 ± 0.26	9.80 ± 0.21	27.64 ± 0.25	29.80 ± 0.30	11.00 ± 0.14	29.80 ± 0.32	32.34 ± 0.36	11.90 ± 0.02	30.75 ± 0.35	35.13 ± 0.36	12.26 ± 0.16	31.14 ± 0.31	36.50 ± 0.34
C18:2 n6t	n.d.	n.d.	n.d.	n.d.	n.d.	n.d.	n.d.	n.d.	n.d.	n.d.	n.d.	n.d.	n.d.	n.d.	n.d.	n.d.	n.d.	n.d.
C18:2 n6	17.96 ± 0.27 ^a^	3.42 ± 0.09 ^b^	3.02 ± 0.10 ^c^	18.19 ± 0.26	12.90 ± 0.11	11.07 ± 0.17	18.23 ± 0.32	13.00 ± 0.18	11.20 ± 0.16	18.45 ± 0.20	13.80 ± 0.20	12.46 ± 0.15	19.10 ± 0.22	15.20 ± 0.21	13.00 ± 0.18	19.16 ± 0.23	16.01 ± 0.17	14.03 ± 0.12
C18:3 n6	1.01 ± 0.12 ^a^	0.07 ± 0.03 ^b^	0.27 ± 0.01 ^c^	1.10 ± 0.03	0.50 ± 0.03	1.89 ± 0.10	1.10 ± 0.02	0.54 ± 0.10	1.90 ± 0.07	1.11 ± 0.05	0.59 ± 0.09	2.00 ± 0.10	1.13 ± 0.04	0.75 ± 0.05	2.50 ± 0.10	1.14 ± 0.08	0.80 ± 0.03	2.60 ± 0.10
C20:1 n9	3.98 ± 0.08 ^a^	n.d.^b^	0.49 ± 0.04 ^c^	4.30 ± 0.07	0.05 ± 0.01	0.78 ± 0.06	4.47 ± 0.08	0.19 ± 0.06	0.82 ± 0.05	5.28 ± 0.16	0.29 ± 0.11	1.68 ± 0.13	5.74 ± 0.20	0.35 ± 0.03	1.90 ± 0.08	6.10 ± 0.10	0.45 ± 0.02	2.33 ± 0.09
C18:3 n3	0.10 ± 0.04 ^b^	0.21 ± 0.08 ^a^	0.26 ± 0.03 ^a^	0.11 ± 0.02	1.27 ± 0.03	0.99 ± 0.06	0.13 ± 0.03	1.28 ± 0.14	1.01 ± 0.12	0.15 ± 0.08	1.30 ± 0.11	1.03 ± 0.10	0.17 ± 0.03	1.40 ± 0.09	1.13 ± 0.03	0.18 ± 0.05	1.58 ± 0.09	1.20 ± 0.08
C20:2 n6	2.83 ± 0.06 ^b^	0.07 ± 0.02 ^a^	0.09 ± 0.01 ^a^	3.00 ± 0.06	1.13 ± 0.05	0.78 ± 0.05	3.01 ± 0.07	1.15 ± 0.03	0.86 ± 0.01	3.08 ± 0.10	1.16 ± 0.06	0.90 ± 0.02	3.10 ± 0.10	1.20 ± 0.11	1.08 ± 0.09	3.12 ± 0.11	1.22 ± 0.07	1.12 ± 0.03
C20:3 n6	2.60 ± 0.03 ^b^	0.25 ± 0.10 ^a^	0.33 ± 0.04 ^a^	2.78 ± 0.03	1.29 ± 0.12	0.65 ± 0.05	2.80 ± 0.05	1.30 ± 0.20	0.68 ± 0.09	2.85 ± 0.07	1.35 ± 0.20	0.70 ± 0.06	3.00 ± 0.10	1.38 ± 0.10	0.88 ± 0.05	3.03 ± 0.04	1.40 ± 0.06	0.90 ± 0.03
C20:4 n6	16.97 ± 0.27 ^b^	0.33 ± 0.13 ^a^	0.38 ± 0.03 ^a^	17.40 ± 0.27	1.97 ± 0.11	1.25 ± 0.10	17.45 ± 0.26	1.98 ± 0.10	1.27 ± 0.09	17.50 ± 0.30	2.04 ± 0.05	1.30 ± 0.10	17.65 ± 0.29	2.09 ± 0.15	1.42 ± 0.09	17.67 ± 0–27	2.10 ± 0.11	1.46 ± 0.09
C24:0	7.00 ± 0.14 ^b^	0.09 ± 0.02 ^a^	0.07 ± 0.01 ^a^	5.98 ± 0.13	0.08 ± 0.04	0.08 ± 0.01	5.73 ± 0.09	0.08 ± 0.03	0.06 ± 0.03	5.07 ± 0.14	0.05 ± 0.01	0.04 ± 0.01	4.36 ± 0.20	0.03 ± 0.01	0.02 ± 0.01	4.11 ± 0.16	0.01 ± 0.01	0.01 ± 0.01
C20:5 n3	1.00 ± 0.05 ^b^	n.d. ^a^	n.d. ^a^	1.08 ± 0.06	0.10 ± 0.05	0.23 ± 0.08	1.08 ± 0.04	0.11 ± 0.05	0.27 ± 0.10	1.10 ± 0.11	0.13 ± 0.05	0.29 ± 0.11	1.13 ± 0.09	0.15 ± 0.06	0.33 ± 0.04	1.15 ± 0.11	0.16 ± 0.05	0.40 ± 0.12
C24:1 n9	0.10 ± 0.04 ^b^	0.02 ± 0.03 ^a^	0.02 ± 0.01 ^a^	0.10 ± 0.03	0.08 ± 0.04	0.07 ± 0.01	0.13 ± 0.05	0.15 ± 0.05	0.08 ± 0.01	0.20 ± 0.06	0.20 ± 0.09	0.10 ± 0.04	0.25 ± 0.05	0.26 ± 0.05	0.19 ± 0.04	0.27 ± 0.08	0.28 ± 0.07	0.25 ± 0.04
C22:6 n3	7.50 ± 0.15 ^b^	0.10 ± 0.05 ^a^	0.13 ± 0.04 ^a^	7.54 ± 0.15	0.54 ± 0.07	0.28 ± 0.10	7.54 ± 0.17	0.54 ± 0.05	0.29 ± 0.12	7.68 ± 0.15	0.55 ± 0.13	0.30 ± 0.11	7.79 ± 0.18	0.60 ± 0.11	0.35 ± 0.09	7.88 ± 0.12	0.62 ± 0.10	0.40 ± 0.12
*Total*																		
SFA	31.95 ± 4.14 ^a^	73.69 ± 0.26 ^b^	68.65 ± 0.20 ^c^	29.62 ± 0.100002	54.95 ± 0.33	52.93 ± 1.17	28.76 ± 0.13	50.34 ± 0.30	51.65 ± 0.12	25.43 ± 0.32	45.32 ± 0.10	46.39 ± 0.11	22.61 ± 0.13	41.45 ± 0.12	41.27 ± 0.11	20.89 ± 0.12	39.32 ± 0.08	37.68 ± 0.11
MUFA	18.08 ± 0.09 ^a^	21.86 ± 0.45 ^b^	26.87 ± 0.12 ^c^	19.18 ± 0.33	25.35 ± 0.44	29.93 ± 0.36	19.90 ± 0.08	29.76 ± 0.10	30.87 ± 0.17	22.65 ± 0.12	33.76 ± 0.15	34.63 ± 0.14	24.32 ± 0.09	35.78 ± 0.12	38.04 ± 0.11	25.78 ± 0.10	36.79 ± 0.26	40.21 ± 0.11
PUFA	49.97 ± 0.12 ^b^	4.45 ± 0.13 ^a^	4.48 ± 0.04 ^a^	51.20 ± 0.11	19.70 ± 0.07	17.14 ± 0.09	51.34 ± 0.12	19.90 ± 0.11	17.48 ± 0.10	51.92 ± 0.13	20.92 ± 0.11	18.98 ± 0.10	53.07 ± 0.13	22.77 ± 0.11	20.69 ± 0.08	53.33 ± 0.13	23.89 ± 0.09	22.11 ± 0.11
SFA/MUFA	1.77 ^a^	3.37 ^b^	2.55 ^c^	1.54	2.17	1.77	1.45	1.69	1.67	1.12	1.34	1.34	0.92	1.16	1.08	0.81	1.07	0.94

^1^ Abbreviations: P, Primiparous; S, Secondiparous; PL, Pluriparous. T0—time point of control diet; T1,T2,T3,T4—time points of experimental diet supplementation; T5—45 days after the last administration of the enriched feed (end lactation); MUFA, monounsaturated fatty acid; SFA, saturated fatty acid; PUFA, polyunsaturated fatty acid. Note: Data are reported as mean ± standard deviation. One-way ANOVA (Newman–Keuls test). a,b,c is the statistical significance (One-way ANOVA). T0 C14:0, C16:0, C18:0, C18:1 n9, C20:1 n9 (P vs. S, PL *** *p* < 0.001; S vs. PL *** *p* < 0.001), C16:1 n7t, C20:2 n6, C20:3 n6, C20:4 n6, C24:0, C20:5 n3, C22:6 n3 (P vs. S, PL *** *p* < 0.001), C16:1 n7 (S vs. P, PL *** *p* < 0.001), C18:2 n6, C18:3 n6 (P vs. S, PL *** *p* < 0.001; S vs. PL * *p* < 0.05), C18:3 n3 (P vs. S * *p* < 0.05; P vs. PL ** *p* < 0.01), C24:1 n9 (P vs. S, PL ** *p* < 0.01). SFA (P vs. S, PL *** *p* < 0.001; S vs. PL * *p* < 0.05). MUFA (P vs. S *** *p* < 0.001; P, S vs. PL *** *p* < 0.001). PUFA (P vs. S, PL *** *p* < 0.001). SFA/MUFA (P vs. S *** *p* < 0.001; P, S vs. PL *** *p* < 0.001).

**Table 2 animals-13-03225-t002:** Milk composition traits in Italian Holstein–Friesian cows during the polyphenol-enriched diet treatment ^1^.

ITEMS	T0			T1			T2			T3			T4			T5		
	P	S	PL	P	S	PL	P	S	PL	P	S	PL	P	S	PL	P	S	PL
Lactose %	4.57 ± 0.05 ^a/b^	4.54 ± 0.04 ^a^	4.65 ± 0.06 ^b^	4.72 ± 0.05	4.68 ± 0.06	4.75 ± 0.07	4.76 ± 0.05	4.72 ± 0.05	4.76 ± 0.03	4.79 ± 0.05	4.75 ± 0.04	4.78 ± 0.05	4.80 ± 0.05	4.82 ± 0.07	4.87 ± 0.04	4.79 ± 0.05 ^a^	4.80 ± 0.05 ^a^	4.89 ± 0.05 ^b^
Protein %	3.22 ± 0.04	3.27 ± 0.03	3.30 ± 0.03	3.30 ± 0.05	3.39 ± 0.05	3.36 ± 0.04	3.45 ± 0.07	3.48 ± 0.03	3.46 ± 0.04	3.48 ± 0.11	3.53 ± 0.05	3.50 ± 0.04	3.50 ± 0.05	3.54 ± 0.04	3.56 ± 0.05	3.50 ± 0.03	3.55 ± 0.04	3.55 ± 0.11
Fat %	4.00 ± 0.05 ^a/b^	4.04 ± 0.04 ^a^	3.93 ± 0.05 ^b^	3.94 ± 0.03	3.97 ± 0.05	3.90 ± 0.04	3.91 ± 0.05	3.94 ± 0.04	3.92 ± 0.03	3.89 ± 0.07	3.89 ± 0.08	3.90 ± 0.04	3.74 ± 0.06	3.75 ± 0.04	3.80 ± 0.05	3.70 ± 0.03	3.73 ± 0.04	3.75 ± 0.07
Somatic cell count (SCC) 10^3^/^mL^	400 ± 50	300 ± 55	350 ± 56	300 ± 50	250 ± 40	300 ± 50	285 ± 50	235 ± 40	278 ± 50	250 ± 40	200 ± 45	250 ± 50	150 ± 40	145 ± 40	186 ± 45	150 ± 50	150 ± 50	190 ± 45

^1^ Note: Values represent the mean ± standard deviation. Abbreviations: P, Primiparous; S, Secondiparous; PL, Pluriparous; SCC, Somatic cell count. One-way ANOVA analysis. a,b is the statistical significance (One-way ANOVA): Lactose T0 (S vs. PL * *p* < 0.05); T5 (P, S vs. PL * *p* < 0.05). Fat T0 (S vs. PL * *p* < 0.05). Two-way ANOVA analysis: Lactose: P (T0 vs. T1 ** *p* < 0.01; T0 vs. T2, T3, T4, T5 *** *p* < 0.001). S (T0 vs. T1 ** *p* < 0.01; T0 vs. T2, T3, T4, T5 *** *p* < 0.001; T1 vs. T4 ** *p* < 0.01; T1 vs. T5 * *p* < 0.05). PL (T0 vs. T2, T3 * *p* < 0.05; T0 vs. T4, T5 *** *p* < 0.001; T1,T2 vs. T4 * *p* < 0.05; T1 vs. T5 ** *p* < 0.01; T2, T3 vs. T5 * *p* < 0.05). Protein: P (T0 vs. T2, T3, T4, T5 *** *p* < 0.001; T1 vs. T2 ** *p* < 0.01; T1 vs. T3, T4, T5 *** *p* < 0.001). S (T0 vs. T1 * *p* < 0.05; T0 vs. T2, T3, T4, T5 *** *p* < 0.001; T1 vs. T3, T4, T5 ** *p* < 0.01). PL (T0 vs. T2 ** *p* < 0.01; T0 vs. T2, T3, T4 *** *p* < 0.001; T1 vs. T3 ** *p* < 0.01; T1 vs. T4, T5 *** *p* < 0.001). Fat: P (T0, T1, T2 vs. T4, T5 *** *p* < 0.001; T3 vs. T4 ** *p* < 0.01; T3 vs. T5 *** *p* < 0.001). S (T0 vs. T3 ** *p* < 0.01; T0, T1,T2 vs. T4, T5 *** *p* < 0.001; T3 vs. T4 ** *p* < 0.01; T3 vs. T5 *** *p* < 0.001). PL (T0 vs. T4 ** *p* < 0.01; T0 vs. T5 *** *p* < 0.001; T1 vs. T5 ** *p* < 0.01; T2 vs. T4 * *p* < 0.05; T2 vs. T5 *** *p* < 0.001; T3 vs. T5 ** *p* < 0.01). SCC: P (T0 vs. T3 * *p* < 0.05; T0, T1 vs. T4, T5 *** *p* < 0.001; T2 vs. T4, T5 ** *p* < 0.01; T3 vs. T4, T5 * *p* < 0.05). S (T0 vs. T3 * *p* < 0.05; T0 vs. T4, T5 *** *p* < 0.001; T1 vs. T4, T5 * *p* < 0.05). PL (T0 vs. T3 * *p* < 0.05; T0 vs.T4, T5 *** *p* < 0.001; T1 vs. T4, T5 * *p* < 0.05).

## Data Availability

All the data are available in the manuscript and Appendix A.

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
