# Peer review of "Effects of Olea europaea L. Polyphenols on the Animal Welfare and Milk Quality in Dairy Cows"

_animals, 2023, doi:10.3390/ani13203225_

Round 1
Reviewer 1 Report
The authors made an excellent effort to link the effects of supplementing diet with Olea eu-25 ropaea L. extract with milk quality and welfare status of lactating Holstein cows. The article is well structured, with a robust discussion to promote the use of phenolic compounds in the diet of dairy cows. The conclusions drawn by the authors are well supported by the results.
Perhaps, a suggestion to improve the article might be that the introductory section feels saturated. It includes too many citations, and some information could be omitted to make the reading more comprehensible. The article is supported by solid conclusions and linked to the in-vogue concept of Circular Economy.
The authors use an appropriate experimental design and statistical analysis to compare the effects of Olea eu-25 ropaea L. extract in primiparous, secondiparous and pluriparous cows, and to compare the fatty acid profile of milk at different times during supplementation.
References are relevant and up to date.
Reviewer 2 Report
Dear authors, it seems to me that this manuscript has some problems that affect the quality of the manuscript. You need to consider your objective and your hypothesis to improve all the manuscript.
Further revision of the language is necessary.
Title: My suggestion is: “Effects of Olea europaea L. polyphenols on the animal welfare and milk quality in dairy cows” – This is just a suggestion to be evaluated by the authors.
Line 16: rewrite it. Change “polyphenols” instead of “powder extract PhenoFeed Dry®”. Add the source of these polyphenols.
Line 17: Remove: “hormonal and” because metabolic include hormonal parameters.
Lines 18-20: Those lines are very similar to the previous lines. Rewrite to improve.
Lines 21-22: This is not necessary. Remove it.
Line 23: Remove “of feed enriched” and “molecules”
Abstract: As I read, more variables appear (Title: welfare; summary: metabolism, hormones; abstract: calving order, antioxidant effect). Maybe it's the writing style. Homogenize the descriptions of the manuscript to avoid these confusions.
It is important to add the statistical design.
Add p-values.
It is important to add numbers to improve the description of the results.
The abstract needs to be improved because the objective, content, and conclusion, are confusingly correlated.
Line 26: Add the number of animals. Also, add more data of the animals.
Line 27: Mid lactation is a very long period, be more specific. Add the days.
Line 28: Describes the specific hormones that were evaluated because the word hormone includes many hormones.
Line 32: “sensitive” and “better” are empiric words. Rewrite it.
Line 35-36: “ensuring an adequate nutritional content” - empiric phrase. Rewrite it.
Line 37: Bioactive or functional? This is an example for the reason I asked you to homogenize the expressions in the text.
Lines 38-39: Very empirical writing style. Rewrite it using scientific styles. The reason is that these empirical words have different definitions depending on the reader. On a scale of 10, for me the highest may be from 5 and perhaps the highest for you is from 8.
Keywords: separate the first keyword in two.
Introduction: Natural substances, natural products, plant origin, etc – many expressions. Try to reduce the number of expressions for the same meaning. E.g. biomolecules, phytocompounds.
The introduction is very generic and repetitive. Rewrite it. Add numbers to improve the introduction and avoid repeating the same idea many times.
Lines 43-80: This text is very repetitive. Rewrite it.
Lines 47-49: According to? Add a reference.
Lines 52-53: These lines are confusing.
Line 73: described previously. Repetitive.
Lines 77-80: In the simple summary you describe that your study is the first, however, here you cited a study.
Lines 84-88: Confusing text to read. Rewrite it.
Lines 96-99: Avoid this type of text. (few studies, first study, non-studies, …)
Lines 99-103: Organize and clearly rewrite it. Add the hypothesis previous to the objective.
Lines 102-103: Did you evaluate the functional and nutraceutical power of milk? If you didn't, this cannot be described here. It can be described as a hypothesis or in future implications after the conclusion, before references.
Material and methods: improve the description of the material and methods. Many methods need to be described in more detail.
This topic should be rewritten following a sequence and bringing together the description of the methods in the same subtopic.
Line 112: It is better to add the mean ± standard deviation for lactation phase day and for body weight (by calving order).
Line 113: Add in detail the statistical design and duration of the study.
Line 115: Describe in detail how the extract was given to the animals.
Line 118: This is a very confusing description.
Line 125-127: every day? Describe the blood collection in one topic and no in two. Also, describe the procedure in detail.
Line 141: Add the name of the hormones.
Line 144: Wait, how? Aren't T3, T4 and estradiol hormones too? Why separate the descriptions?
Line 146: Here and through the text. Describe the full name of the items the first time that are used (abbreviation in parentheses) and only the abbreviation thereafter.
Line 151: describe better how is a bi-weekly basis.
Line 157: Whose methodology is this?
Line 178: Lysozyme was described before. Use only one subtopic to describe a specifically item.
Line 190: Isn't it an F test to compare only two variables?
Lines 189-201: Many variables were studied as treatments according to this description. This is not clear in the M&M issue. Better describe the statistical treatments and statistical design in the M&M topic.
Line 202: I read the results and found no simplicity. In the sense that it was difficult to see the effects only in the figures. Add strategic tables with all the data.
Results: I know the authors tried to display the results dynamically; however, the numbers are necessary to understand the results and observe real changes. You can maintain the figures. Use strategic tables (maybe only with means) with all the data.
Discussion: Your discussion is a good review; However, the topic of discussion is to explain their results through biological, physiological, metabolic, etc. approaches, which led them to obtain those results. In this sense, add the discussion of your data.
Lines 384-393: The ideas in this paragraph are good; however, the ideas are not directly correlated. Rewrite it.
Lines 394-400: This is not necessary. Remove it.
Lines 406-412: This is a good description; However, how does this text help the discussion of your data? For me, this is just the description of the results of other authors.
Line 416: Do you evaluate the early lactation phase? If not, add a reference.
Lines 416-418: And what is the explanation for this in your results? Why is this important or not for animals?
Conclusion: Your conclusion is very long. Reduce it. Also, add the subtopic “Future Implications” where you can add some paragraphs of the conclusion.
Reviewer 3 Report
The research, titled 'Effects of Supplementation of Diet with Olea europaea L. Polyphenols on Animal Welfare and Milk Quality in Italian Holstein-Friesian Dairy Cows,' addresses an important and timely topic. I found the subject matter of the article fascinating and read the manuscript with great interest. The paper aligns well with the scope of the journal. However, I believe that in its current form, it has several shortcomings:
- I suggest rewriting the simple summary. According to the author's guidelines, this section should summarize and contextualize your paper within the existing literature in your field. It should be written without technical language or nonstandard acronyms, with the goal of conveying the meaning and importance of this research to non-experts.
- I recommend rewriting the abstract and including more results and the significance of the obtained data.
- The discussion section should be expanded to include practical applications and the study's limitations.
Specific comments:
Lines 43-44: please consider to cite: 10.3390/ani13050797
Lines 72-73: please consider to cite: 10.1016/j.rvsc.2023.03.008
Line 187: Could you please clarify whether you conducted tests for normality and homogeneity on your data before proceeding with the statistical analysis? It's crucial to ensure that the assumptions underlying your chosen statistical methods are met. I recommend referring to the guidelines outlined in [proposed reference, e.g., 10.1080/1828051X.2020.1827990] for conducting such tests to maintain the rigor and reliability of your analysis.
Reviewer 4 Report
GENERAL COMMENTS
In the study animals-2611711 entitled “Effects of supplementation of diet with Olea europaea L. polyphenols on the animal welfare and milk quality in Italian Holstein-Friesian dairy cows” the authors have evaluated the effect of dietary supplementation with Olea europaea L. extract on the animal welfare and milk quality of lactating Italian Holstein-Friesian dairy cows at three different parity classes. The manuscript presents interesting data and that the topic of the manuscript is of significant interest and appropriate for the Journal. The manuscript is written in an acceptable English language. The presentation and the length of the manuscript are adequate as the description of the experimental plan. In particular, the title accurately reflects the major findings of the work; the keywords represent the article adequately and the abstract section well summarizes the background, methodology, results, and significance of the study; the introduction section is well written but can be improved, the topic of the study is well stated and supported by adequate bibliographic information; material and methods section is well written, adequately and meticulously describes the methods applied in the study; results section is clear and the obtained findings were well explained. The discussion and conclusion sections are not clear. Many paragraphs are too long and unclear, please rewrote these sections. However, some major changes I suggest throughout the text. In view of this, I believe that the manuscript is suitable for publication after MAJOR REVISION.
SPECIFIC COMMENTS:
The title accurately reflect the major findings of the work.
The abstract clearly summarize the background, methodology, results, and significance of the study. However, AA should improve some sentences.
The introductory section is well written; however, it may be reduced.
The section of Materials and Methods is clear for the reader, but the authors should check this section and correct many punctuation errors.
Please change “calving orders” with “parity classes” during the text (lines 100, 112, etc…)
Please rewrite discussion and conclusions of the study.
Tables and figures are generally good, and they well represent results obtained. Data in Tables were not duplicated in the text.
The reference list should be improved. Correct and uniform the references during the text to the journal style.
Round 2
Reviewer 2 Report
Dear authors, it seems to me that this manuscript has some issues that affect the quality of the manuscript. Please, improve the manuscript based on the suggestions.
(Abstract) Dear author, statistical design refers to experimental design (Latin square, randomized design, block design, etc.), the use of a factorial scheme, etc. Lactation days, number of animals, calving order and others are important data to support your statistical design.
Line 86: Remove this line is unnecessary.
Lines 87-89: I think the authors tried to write a more complex objective in this part and this is complex and interesting to read; however, when I read this objective, the correlation of this objective with the title and objective of the abstract is different (Lactation phase and parity class). Rewrite the objective in the abstract.
Line 100: Add the statistical design as recommended above for the abstract.
The treatments are quantitative; why was polynomial contrast or regression analysis not used? This type of analysis would show what level of additive inclusion promoted an improvement in performance and product quality.
“We have emphasized that not only extracts from the olive-oil supply chain but also extracts from other agri-food chains have antioxidant effects and do not alter metabolic parameters in dairy cows”:
I understand your response, but its purpose is not to discuss other agri-foods, but to discuss the olive oil that you used in animal feed.
Additionally, I understand that data from other authors is used to support the data; however, that is not a discussion, it is just a corroboration. A discussion is to explain the results. If other authors do not explain their results, it has nothing to do with their results or their study. Independently, you need to write your theories or hypotheses explaining your results.
In this sense, improve your discussion.
Read with me an example similar to your conclusion: global temperature is increasing, it is necessary to use umbrellas because it reduces the effects of rising temperatures.
The above description is actually very generic. In that sense, it is necessary for you to improve your conclusion. In the conclusion it is necessary to describe what level of inclusion promoted better results. How much did the results improve?, etc. As it stands, your conclusion is very generic. Furthermore, the conclusion must correlate with the objective.
Reviewer 3 Report
Dear Authors,
I wanted to extend my heartfelt congratulations to you and your team for the outstanding job you've done in revising your paper. I am genuinely impressed by the way you have meticulously incorporated the suggested revisions. Your commitment to improving the article's quality is evident, and I must say that the final result is nothing short of exceptional. The transformation from the initial draft to the current version is remarkable and a testament to your dedication to excellence.
Author Response
We would like to really thank the reviewer for the constructive criticisms.
Reviewer 4 Report
The manuscript animals-2611711 entitled “Effects of supplementation of diet with Olea europaea L. polyphenols on the animal welfare and milk quality in Italian Holstein-Friesian dairy cows” is is now suitable for publication.
Author Response
We would like to really thank the reviewer for the constructive suggestions.
Round 3
Reviewer 2 Report
Dear authors, it seems to me that this manuscript has relevance in the scientific world. I think the authors made the suggested changes, plus they considered the best writing options for a quality manuscript; therefore, I recommend approval.